# ARBOREAL NEURAL NETWORK

## ABSTRACT

Connectionist models and symbolic models have long embodied two divergent paradigms: the former excel at differentiable representation learning yet struggle with transparency, while the latter deliver explicit rule-based reasoning but resist gradient-based optimization. We introduce Arboreal Neural Networks (ArbNN), a neural–symbolic framework that unifies these paradigms both computationally and conceptually. At the design level, ArbNN departs fundamentally from prior neuralized-tree models through a depth-aware routing mechanism and a topology-informed softmax aggregation, which together enable one-shot multi-path gradient propagation and consequently achieving rapid and well-conditioned optimization dynamics and high parallel inference efficiency. At the conceptual level, ArbNN reveals that decision-tree branching and self-attention routing are two realizations of the same conditional computation primitive. We prove a structural isomorphism between a decision tree and a single-query attention head, enabling a differentiable architecture that faithfully preserves symbolic decision logic. A key property of ArbNN is Bidirectional Fidelity, ensuring that the neural module can be compiled from—and losslessly decompiled back into—a symbolic tree, yielding both ordering consistency in ranking behavior and explicit, auditable interpretability via reconstructed if–else rules. ArbNN further supports GBDT-based initialization, allowing it to inherit strong inductive biases and integrate seamlessly with existing production workflows. Empirically, ArbNN achieves state-of-the-art performance on various public tabular benchmarks and delivers consistent gains under temporal distribution shift in large-scale industrial credit-risk systems. To support realistic evaluation, we additionally contribute TabCredit, a feature-rich, temporally partitioned dataset built from millions of real-world loan applications. Together, these results demonstrate that ArbNN forms a unified, reversible, and practically deployable bridge between symbolic reasoning and neural computation for high-stakes tabular domains.

## 1 INTRODUCTION

A prevailing challenge in unifying symbolic Chen & Guestrin (2016); Ke et al. (2017); Prokhorenkova et al. (2018) and neural paradigms Lecun et al. (1998); Krizhevsky et al. (2012); He et al. (2016); Vaswani et al. (2017); Radford et al. (2018); Devlin et al. (2019) lies not merely in hybridizing tree-like components with neural modules, but in determining whether they share a common conditional computation primitive. Prior neural–symbolic approaches often soften discrete decision boundaries, approximate symbolic operators with differentiable surrogates, or impose post-hoc explanations on neural networks. While such techniques yield practical utility, they do not bridge the core conceptual divide: the lack of a principled and invertible correspondence between discrete symbolic structures and continuous neural representations. Recent work in modular and interpretable neural systems further highlights this difficulty. Typical studies on weight-sparse Transformers Gao et al. (2025) and modular manifolds Bernstein (2025) demonstrate that learning sparse, modular, or symbolic structure *from scratch* remains intrinsically challenging. Neural activations tend to remain entangled and polysemantic, even under strong regularization, and enforcing routing, modularity, or discreteness during training is often unstable and rarely yields audit-grade interpretability.

We introduce Arboreal Neural Networks (ArbNN), a neural architecture that provides a bidirectional correspondence between decision trees and self-attention. A decision tree can be compiled into a differentiable module that is structurally homomorphic to a single-query attention head, and

the resulting module can be decompiled back into an equivalent symbolic tree with no loss of interpretability. This establishes attention-based routing and symbolic branching as two forms of the same conditional computation primitive, enabling ArbNN to combine global, auditably symbolic reasoning with end-to-end trainability and competitive function approximation. ArbNN employs a more principled and informative initialization that uses a pretrained GBDTs as a structural sparsity template, rather than relying on a purely random sparse topology as in weight-sparse Transformer regimes Gao et al. (2025). This design provides a meaningful and domain-aligned structural prior, following the broader shift toward fixed-structure models with differentiable parameter refinement. More realistically for real-world deployments, it also inherits the most widely used tree models, avoiding migration cost or performance risk and providing a stable foundation for fine-tuning.

We also contribute an industrial dataset. As is well-known, most academic tabular benchmarks adopt random splits to construct their datasets, with the aim of evaluating generalization under the assumption of Independent and Identically Distributed (I.I.D.) data. However, these benchmarks often overlook the temporal drift and strong time dependencies inherent in real-world applications. Additionally, industrial datasets are typically created through complex acquisition processes and extensive feature engineering, leading to numerous informative variables. Despite their value, such datasets are rarely publicly available Rubachev et al. (2025). To address this gap, we present **TabCredit**, a large-scale, feature-rich credit risk dataset derived from real-world loan applications and organized with temporal splits. TabCredit enables benchmarking under realistic conditions of temporal drift and complex feature engineering—two critical aspects that are largely absent from existing public datasets.

Our main contributions are summarized as follows:

- We present **Arboreal Neural Networks (ArbNN)**, a neural–symbolic framework built around *bidirectional fidelity*: the model can be compiled from and decompiled back into a symbolic tree while preserving ordering-consistent ranking behavior and explicit, auditable interpretability. ArbNN further unify symbolic decision-tree reasoning with connectionist representation learning in a single differentiable architecture.

- We introduce **ArborCell**, the differentiable computation unit of ArbNN. ArborCell employs a depth-aware routing mechanism together with a topology-informed softmax aggregation, enabling one-shot multi-path computation for efficient training and parallel inference. We additionally prove that ArborCell is *structurally homomorphic* to a single-query self-attention head, establishing the formal correspondence between symbolic branching and neural routing.

- Our model achieves State-of-the-Art (SOTA) performance across various tabular prediction datasets, outperforming both traditional ensemble trees and existing neural models. In addition, it has been successfully deployed in real-world commercial risk control scenarios involving large-scale credit flow, demonstrating its practical efficacy.

- We introduce **TabCredit**, a large-scale and feature-rich dataset for credit risk assessment, constructed from real-world loan applications. The dataset comprises 2 million samples with 100 anonymized, engineered features and adheres to strict in-time and Out-of-Time (OOT) splits.

## 2 RELATED WORK

### 2.1 TRADITIONAL TREE-BASED MODELS

Traditional tree-based models, such as XGBoost Chen & Guestrin (2016), LightGBM Ke et al. (2017), and CatBoost Prokhorenkova et al. (2018), excel in handling tabular data. By constructing hierarchical feature routing systems based on split points determined from statistical criteria like information gain, they align well with the localized dependencies and additive interactions common in such data. However, this reliance on global statistics can cause the models to overlook challenging or underrepresented instances, thereby constraining their effectiveness. Recent work has also emphasized the importance of robust hyperparameter defaults. Holzmüller et al. (2024) demonstrates that carefully pre-tuned configurations for GBDTs consistently outperform standard defaults across diverse tabular tasks. In parallel, neural architectures offer an alternative by leveraging end-to-end optimization of

all parameters, which enhances their ability to capture complex data distributions and reduces the risk of neglecting rare or difficult samples.

## 2.2 DIFFERENTIABLE NEURAL ARCHITECTURES

End-to-end differentiable architectures have been widely explored for tabular data. TabTransformer Huang et al. (2020) is an early work of applying self-attention to embeddings of categorical features within tabular data, enabling context-aware feature encoding. FT-Transformer Gorishniy et al. (2021) expanded on TabTransformer by jointly processing numerical and categorical features through a unified Transformer block. TFWT Zhang et al. (2024) introduced dynamic feature weighting within Transformer layers to highlight context-dependent variable importance. SwitchTab Wu et al. (2024) employed an asymmetric encoder-decoder structure to enhance feature representations and interpretability. Trompt Chen et al. (2024b) utilized prompt-based learning to separate global schema information from instance-specific noise. TabM Gorishniy et al. (2025) Achieves ensemble benefits for tabular deep learning by combining a shared MLP backbone with lightweight adapters for parameter-efficient diversity. TabPFN Hollmann et al. (2025) is a pre-trained tabular foundation model that, by learning to perform Bayesian-style inference from diverse synthetic priors, produces accurate zero- or few-shot predictions on small-to-medium tabular datasets while delivering orders-of-magnitude faster inference than tuned AutoML baselines

Despite their adaptability in handling diverse tabular data, differentiable neural tabular models often struggle to consistently outperform tree ensembles across many tasks—particularly in scenarios with limited data or heavily engineered features Rubachev et al. (2025). Moreover, these models lack the intuitive interpretability inherent to tree-based models.

## 2.3 HYBRID NEURAL-TREE ARCHITECTURES

Hybrid neural-tree architectures differ mainly in how decision-tree logic is integrated with neural computation. Early approaches implement *independent soft trees*, where the predictive model itself is structured as a differentiable ensemble. These methods replace hard if–else splits with *soft gating* (sigmoid or softmax) and compute predictions as mixtures over leaves via *path probabilities*. Representative examples include Neural Decision Forests Kontschieder et al. (2015) and Soft Decision Trees Frosst & Hinton (2017), which use per-node binary soft splits (sigmoid-based left/right routing); Neural Random Forests Biau et al. (2019) implement a depth-agnostic routing mechanism in which each internal node computes a signed matrix product over input features, producing a single-leaf activation that differentiably mimics the hard path selection of classical decision trees. NODE Popov et al. (2020), which performs layerwise oblivious soft splits and GRANDE Marton et al. (2024), which combines softmax-based feature selection with sigmoid-based thresholding to stabilize gradient flow in deep tree ensembles. Another direction embeds tree-based reasoning within neural networks as specialized modules, such as the Adaptive Neural Trees Tanno et al. (2019), which insert learnable neural Transformers along paths and dynamically grow or prune the tree to enable adaptive structure learning; Tree Ensemble Layer (TEL) Hazimeh et al. (2020), which integrates differentiable tree ensembles as conditional-computation layers; Adaptive Neural Trees Tanno et al. (2019), which insert learnable neural Transformers along paths and dynamically grow or prune the tree to enable adaptive structure learning; Net-DNF Katzir et al. (2021), which encodes symbolic decision logic in disjunctive normal form as neural layers, preserving rule-based reasoning; and NCART Luo & Xu (2024), which incorporates differentiable oblivious trees into residual networks to improve both efficiency and interpretability. Despite these advances, existing soft-tree models still rely on **learned dense gating weights**, which eliminate structural explicitness and node-level interpretability—their explanations are limited to feature-importance scores rather than explicit decision logic. Furthermore, these models depend on **recursive path-probability computation**, leading to unstable gradients and inefficient training.

In contrast to existing soft-tree models, which rely on dense gating functions and recursive path-probability computations, our methods adopts a structurally grounded realization of tree computation. ArbNN first encodes the tree topology through a depth-aware routing matrix $\mathbf{P}$ that assigns exponentially decayed influence to higher-level splits, preserving the hierarchical semantics of the original tree. A topology-informed softmax aggregation $\boldsymbol{\alpha}$ then produces a continuous distribution over leaves that the fully matched leaf attains the maximal activation, the exponential attenuation ensures graded

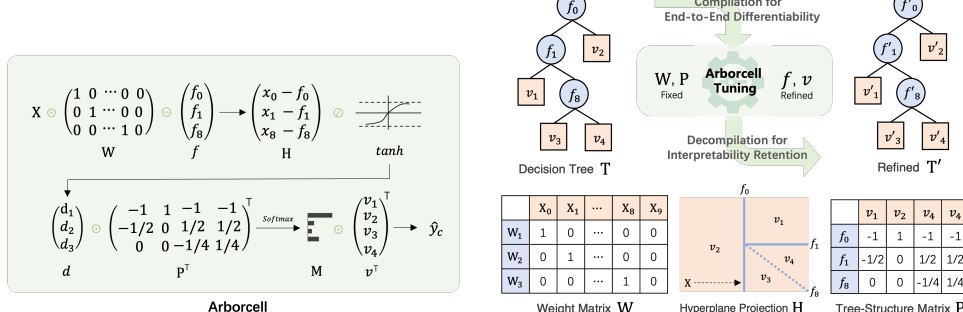

Figure 1: Arboreal Neural Network transforms a decision tree $\mathbf{T}$ into an ArborCell, enabling **end-to-end differentiability** while retaining the symbolic structure for **interpretability**. The left-green panel illustrates the compilation of tree $\mathbf{T}$ (via the Algorithm in Appendix A into its ArborCell representation. Matrix $\mathbf{W}$ encodes feature–node assignments, and matrix $\mathbf{P}$ encodes the tree topology. Together with the bias vector $f$ (split thresholds), these define the hyperplane projections that relate input $\mathbf{X}$ to the internal decision nodes. After training, the ArborCell can be decompiled back into a refined tree $\mathbf{T}'$, where $(\mathbf{W}, \mathbf{P})$ remain fixed and only $(f, v)$ are fine-tuned, thereby improving predictive performance while preserving the original tree structure.

activation for near-miss paths, enabling gradient flow through all structurally compatible paths. These components are combined into a **one-shot multi-path formulation** that replaces recursive routing with a single matrix-based computation, yielding numerically stable and highly parallel inference. Expressing the routing and aggregation operations in this matrix form reveals that the resulting module is *structurally homomorphic* to a single-query self-attention head: internal nodes play the role of queries, the topology matrix defines keys, and leaf predictions serve as values. This homomorphism provides a principled bridge between symbolic branching and neural routing, allowing ArbNN to retain full symbolic interpretability while supporting end-to-end differentiable refinement and exact post-training reversibility to decision-tree form.

## 3 METHOD

### 3.1 ARBOREAL NEURAL NETWORK

Arboreal Neural Network is a wide, shallow, and differentiable neural architecture that reinterprets the discrete logic of tree-based models. Each component, called an ArborCell, functions similarly to a tree in tree-based models. The overall prediction of the ArbNN model is obtained by aggregating the logits from its individual ArborCells:

$$\hat{y}_{\text{ArbNN}} = \sum_{c=1}^{C} \hat{y}_c \tag{1}$$

where $C$ denotes the total number of ArborCells in the ArbNN model. The ArbNN model is initially configured using a pretrained decision tree ensemble (e.g., XGBoost), which provides critical elements, including split conditions, leaf node weights, and structural topology.

### 3.2 SPARSE NEURAL TREE: ARBORCELL

As illustrated in Figure 1, each ArborCell processes a sample $\mathbf{X} \in \mathbb{R}^t$ and produces a prediction $\hat{y}_c$. This process emulates tree prediction logic through a series of neural computations, including matrix multiplication and activation functions. The necessary matrices are derived from a decision tree, denoted as $\mathbf{T}$, using Algorithm in Appendix A. Specifically, we extract the sparse weight matrix $\mathbf{W}$ that indicates feature-node relationships, a bias vector $f$ that represents split points, a tree-structure matrix $\mathbf{P}$ that reflects the hierarchical subspace organization, and leaf node weights $v$.

**Weight Matrix** $\mathbf{W} \in \mathbb{R}^{n_i \times t}$ is a sparse one-hot matrix, where each row corresponds to an internal node, and the non-zero entry in row $i$ indicates the index of the feature selected by the $i$-th internal node from a set of $t$ candidate features. $n_i$ is the total number of inner nodes.

**Tree-structure Matrix** $\mathbf{P} \in \mathbb{R}^{n_i \times n_j}$ captures the tree's structure. Each entry $p_{ij}$ indicates whether the leaf node $j$ is reachable from the internal node $i$ and on which side of the hyperplane the leaf lies. Specifically, for each pair of nodes, the value of $p_{ij}$ is defined as:

$$
p_{ij} = \begin{cases} \left(\frac{1}{2}\right)^{dep_j - 1}, & \text{if node } j \in \text{Desc}_{\text{L}}(i) \\ -\left(\frac{1}{2}\right)^{dep_j - 1}, & \text{if node } j \in \text{Desc}_{\text{R}}(i) \\ 0, & \text{if node } j \notin \text{Desc}(i) \end{cases} \tag{2}
$$

where $dep_j$ represents the number of edges from the root to leaf node $j$ and $\text{Desc}_{\text{L}}(i)$ and $\text{Desc}_{\text{R}}(i)$ denote the sets of leaf nodes that lie on the left or right subtrees of node $i$, respectively. The sign indicates whether a leaf is on the left or right of the hyperplane ($+1$ for left and $-1$ for right) defined by node $i$, and the magnitude is weighted by $(1/2)^{dep_j - 1}$. A value of 0 indicates that the leaf is not reachable from node $i$. We adopt an exponentially decaying weighting design for the matrix $\mathbf{P}$ (with base $1/2$) to emulate the hierarchical subspace selection process of decision trees in a fully end-to-end learnable manner. By assigning exponentially scaled weights to nodes closer to the root, we effectively mitigate the imbalances in deeply skewed trees, ensuring robust path selection through matrix operations.

### 3.3 Feed Forward Computation in ArborCell

**Hyperplane Projection H**: Each internal node defines a hyperplane in the feature space, represented by a weight-bias pair $(\mathbf{W}_i, f_i)$. To quantify the relationship of $\mathbf{X}$ to this hyperplane, we define the *hyperplane projection*:

$$
\mathbf{H}_i = \mathbf{W}_i \mathbf{X} - f_i \tag{3}
$$

Here, $\mathbf{H}_i$ represents the signed distances of $\mathbf{X}$ to the hyperplane. By evaluating $\mathbf{X}$ against $n_i$ learned hyperplanes and form $\mathbf{H} \in \mathbb{R}^{n_i \times 1}$.

**Decision Vector d**: The hyperplane projections are passed through a scaled hyperbolic tangent function to obtain the decision vector $\mathbf{d} \in \mathbb{R}^{n_i}$:

$$
\mathbf{d}_i = \tanh(\tau_1 \cdot \mathbf{H}_i) \tag{4}
$$

where $\tau_1 > 0$ is a temperature parameter that modulates the sharpness of the activation function. A higher $\tau_1$ pushes $\mathbf{d}_i$ closer to the extremal values $-1$ and $+1$, thereby approximating the hard binary decisions of traditional trees.

In this way, each element $\mathbf{d}_i$ reflects the direction and confidence of the input sample with respect to the $i$-th decision boundary: $d_i \approx +1$ indicates a confident right-child traversal, $d_i \approx -1$ indicates a left-child preference, and intermediate values arise when $\tau_1$ is small or the input lies near the hyperplane. As a whole, the vector $\mathbf{d}$ serves as a differentiable analogue of the discrete split routing in decision trees, encoding soft decisions over internal nodes.

**Subspace Affinity Vector M**: The decision vector $\mathbf{d} \in \mathbb{R}^{n_i}$ is linearly transformed by the tree-structure matrix $\mathbf{P} \in \mathbb{R}^{n_i \times n_j}$ to obtain the subspace affinity vector:

$$
\mathbf{M} = \mathbf{P}^\top \mathbf{d} \in \mathbb{R}^{n_j} \tag{5}
$$

where $n_j$ denotes the number of leaf nodes. Each element $\mathbf{M}_j$ reflects the aggregated directional alignment of the input with respect to the internal nodes along the path to leaf node $j$, incorporating both routing consistency and hierarchical depth weighting. This process can be viewed as a soft generalization of decision path traversal in conventional trees.

To obtain a normalized distribution over leaf subspaces, the subspace affinity vector $\mathbf{M}$ is passed through a softmax function: $\boldsymbol{\alpha} = \text{Softmax}(\tau_2 \cdot \mathbf{M})$, where the temperature parameter $\tau_2 > 0$ controls the peakedness of the distribution. The resulting vector $\boldsymbol{\alpha} \in \mathbb{R}^{n_j}$ encodes a probabilistic allocation of the input across all leaf regions, enabling smooth, differentiable transitions between them.

| Model | Classification ↑ | | | | | | | Regression ↓ | | | |
|---|---|---|---|---|---|---|---|---|---|---|---|
| | MA | H16 | CR | JA | DI | MI | CA | YP | HO | CH | ME |
| MLP Haykin (1994) | 0.938 | 0.940 | 0.830 | 0.869 | 0.650 | 0.978 | 0.953 | 0.881 | 0.435 | 0.467 | 0.140 |
| ResNet He et al. (2016) | 0.938 | 0.946 | 0.828 | 0.871 | 0.649 | 0.983 | 0.960 | 0.875 | 0.434 | 0.468 | 0.147 |
| FTTransformerBucket Gorishniy et al. (2021) | 0.939 | 0.950 | 0.857 | 0.866 | 0.649 | 0.986 | 0.960 | 0.885 | 0.441 | 0.468 | 0.147 |
| FTTransformer Gorishniy et al. (2021) | 0.936 | 0.949 | 0.828 | 0.871 | 0.652 | 0.981 | 0.959 | 0.896 | 0.445 | 0.461 | 0.141 |
| TabNet Arik & Pfister (2021) | 0.937 | 0.936 | 0.828 | 0.862 | 0.650 | 0.977 | 0.944 | 0.885 | 0.451 | 0.544 | 0.140 |
| TabTransformer Huang et al. (2020) | 0.918 | 0.937 | 0.816 | 0.867 | 0.642 | 0.980 | 0.928 | 0.886 | 0.539 | 0.545 | 0.141 |
| Trompt Chen et al. (2024b) | 0.942 | 0.952 | 0.829 | 0.882 | 0.652 | 0.982 | 0.964 | 0.874 | 0.418 | 0.424 | 0.140 |
| GRANDE Marton et al. (2024) | 0.939 | 0.945 | 0.833 | 0.871 | 0.653 | 0.982 | 0.959 | 0.883 | 0.432 | 0.460 | 0.141 |
| ExcelFormer Chen et al. (2024a) | 0.939 | 0.950 | 0.833 | 0.879 | 0.651 | 0.982 | 0.969 | 0.890 | 0.411 | 0.431 | 0.142 |
| SwitchTab Wu et al. (2024) | 0.939 | 0.948 | 0.828 | 0.872 | 0.651 | 0.981 | 0.961 | 0.880 | 0.430 | 0.452 | 0.144 |
| TabPFN Hollmann et al. (2025) | 0.939 | 0.946 | 0.854 | 0.873 | 0.638 | 0.980 | 0.967 | 0.888 | 0.400 | 0.452 | 0.143 |
| TabM Gorishniy et al. (2025) | 0.943 | 0.947 | 0.843 | 0.875 | 0.651 | 0.982 | 0.963 | 0.877 | 0.396 | 0.435 | 0.142 |
| CatBoost Prokhorenkova et al. (2018) | 0.924 | 0.948 | 0.861 | 0.863 | 0.649 | 0.986 | 0.963 | 0.876 | 0.443 | 0.439 | 0.145 |
| LightGBM Ke et al. (2017) | 0.943 | 0.951 | 0.862 | 0.881 | 0.652 | 0.986 | 0.972 | 0.863 | 0.384 | **0.395** | 0.141 |
| XGBoost Chen & Guestrin (2016) | 0.947 | 0.951 | 0.862 | 0.868 | 0.653 | 0.986 | 0.966 | 0.857 | 0.401 | 0.418 | 0.140 |
| ArbNN | **0.974** | **0.987** | **0.884** | **0.923** | **0.657** | **0.992** | **0.986** | **0.832** | **0.359** | 0.406 | **0.130** |

Table 1: Comparisons on public datasets, with the highest performance for each dataset highlighted in **bold**. Experimental results are averaged over three independent runs. The datasets are divided into two groups, one for classification, evaluated using the AUC-ROC metric, and one for regression, assessed using RMSE metric.

**ArborCell Prediction** $\hat{y}_c$: Let $\mathbf{v} \in \mathbb{R}^{n_j}$ denote the prediction values associated with leaf nodes. The final output of the ArborCell is given by the expectation over leaf predictions:

$$\hat{y}_c = \sum_{j=1}^{n_j} \alpha_j v_j \tag{6}$$

This output represents a soft aggregation of all leaf values, weighted by how consistently the input aligns with the decision paths encoded by the internal node activations and their hierarchical routing.

## 3.4 STRUCTURAL HOMOMORPHISM TO SELF-ATTENTION

We show that the computation of an *ArborCell*—the differentiable counterpart of a decision tree unit—is structurally homomorphic to a single-query self-attention head. We retain the notation from the Tree Parsing Algorithm (Appendix 1).

**Theorem.** An ArborCell is structurally homomorphic to a single-query self-attention head in the following sense.

*(1)* Each internal node $i \in \{1, \ldots, n_i\}$ contributes one coordinate to the decision vector $\mathbf{d} \in \mathbb{R}^{n_i}$, i.e., one axis of the query embedding space. Hence the number of internal nodes $n_i$ equals the attention *embedding dimension*.

*(2)* Each leaf $\ell \in \{1, \ldots, n_j\}$ is associated with (a) a path encoding given by the $\ell$-th column of the tree-structure matrix $\mathbf{P} \in \mathbb{R}^{n_i \times n_j}$ and (b) a scalar leaf value $v_\ell$ from $\mathbf{v} \in \mathbb{R}^{n_j}$. Thus the number of leaves $n_j$ equals the *sequence length* (number of tokens).

Under the identification

$$\mathbf{Q}_{\text{(CLS)}} = \mathbf{d}^\top \in \mathbb{R}^{1 \times n_i}, \qquad \mathbf{K}_{\text{(tokens)}} = \mathbf{P}^\top \in \mathbb{R}^{n_j \times n_i}, \qquad \mathbf{V}_{\text{(tokens)}} = \mathbf{v} \in \mathbb{R}^{n_j \times 1} \tag{7}$$

the ArborCell output matches the single-query attention form:

$$\text{ArborCell}(\mathbf{d}, \mathbf{P}, \mathbf{v}) = \text{Attention}\big(\mathbf{Q}_{\text{(CLS)}}, \mathbf{K}_{\text{(tokens)}}, \mathbf{V}_{\text{(tokens)}}\big) \tag{8}$$

**Proof.**

*(1) Subspace affinity.* By the ArborCell definition (Sec. 3.2), the subspace affinity vector is $\mathbf{M} = \mathbf{P}^\top \mathbf{d} \in \mathbb{R}^{n_j}$. Using the above identification, this is equivalently

$$\mathbf{M} = \mathbf{K}_{\text{(tokens)}} \mathbf{Q}_{\text{(CLS)}}^\top \in \mathbb{R}^{n_j \times 1} \tag{9}$$

which is precisely the key–query affinity in self-attention.

*(2) Normalization.* Routing weights over leaves are obtained by normalizing the affinities:

$$\boldsymbol{\alpha} = \text{softmax}\big(\tau_2 \mathbf{K}_{\text{(tokens)}} \mathbf{Q}_{\text{(CLS)}}^\top\big) \in \mathbb{R}^{n_j} \tag{10}$$

| Model | MOB2 | MOB3 | MOB4 | MOB5 | MOB6 | MOB7 | MOB8 | MOB9 | MOB10 | MOB11 | MOB12 |
|---|---|---|---|---|---|---|---|---|---|---|---|
| MLP Haykin (1994) | 54.73 | 52.19 | 50.35 | 48.92 | 47.24 | 45.68 | 43.91 | 42.49 | 41.26 | 39.99 | 38.82 |
| ResNet He et al. (2016) | 55.39 | 52.89 | 50.94 | 49.39 | 47.57 | 45.90 | 44.06 | 42.70 | 41.48 | 40.20 | 38.96 |
| FTTransformer Gorishniy et al. (2021) | 55.77 | 53.36 | 51.32 | 49.57 | 47.70 | 46.03 | 44.12 | 42.67 | 41.38 | 40.04 | 38.80 |
| TabTransformer Huang et al. (2020) | 51.25 | 48.78 | 47.11 | 45.40 | 43.67 | 42.11 | 40.51 | 39.22 | 38.08 | 36.91 | 35.74 |
| FTTransformerBucket Gorishniy et al. (2021) | 55.80 | 53.42 | 51.35 | 49.61 | 47.75 | 46.09 | 44.16 | 42.71 | 41.43 | 40.07 | 38.85 |
| Trompt Chen et al. (2024b) | 55.54 | 52.88 | 50.99 | 49.22 | 47.34 | 45.68 | 43.85 | 42.37 | 41.05 | 39.72 | 38.46 |
| GRANDE Marton et al. (2024) | 55.49 | 52.80 | 50.91 | 49.13 | 47.24 | 45.59 | 43.72 | 42.22 | 40.89 | 39.64 | 38.38 |
| ExcelFormer Chen et al. (2024a) | 53.80 | 51.69 | 49.74 | 48.36 | 46.77 | 45.34 | 43.71 | 42.40 | 41.22 | 40.01 | 38.79 |
| SwitchTab Wu et al. (2024) | 55.57 | 53.12 | 51.11 | 49.46 | 47.61 | 45.91 | 44.02 | 42.53 | 41.21 | 39.85 | 38.67 |
| TabPFN Hollmann et al. (2025) | 55.51 | 53.01 | 51.06 | 49.28 | 47.48 | 45.84 | 43.90 | 42.48 | 41.17 | 39.85 | 38.61 |
| TabM Gorishniy et al. (2025) | 55.84 | 53.47 | 51.43 | 49.57 | 47.84 | 46.19 | 44.26 | 42.75 | 41.49 | 40.13 | 39.01 |
| XGBoost Chen & Guestrin (2016) | 56.52 | 53.92 | 51.90 | 50.13 | 48.19 | 46.40 | 44.50 | 43.03 | 41.72 | 40.40 | 39.13 |
| LightGBM Ke et al. (2017) | 56.31 | 53.72 | 51.65 | 49.90 | 48.01 | 46.18 | 44.35 | 42.78 | 41.55 | 40.21 | 38.95 |
| **ArbNN** | **57.14** | **54.13** | **52.08** | **50.33** | **48.43** | **46.65** | **44.76** | **43.28** | **41.95** | **40.60** | **39.32** |

Table 2: KS (%) across M2+@MOBN in vintage analysis on the TabCredit Dataset for tracking delinquency trends (M2+: delinquency beyond 31 days; MOBN: months-on-book). The best values are in bold.

mirroring attention-weight normalization. (standard $1/\sqrt{n_i}$ scaling can be incorporated into $\tau_2$ without affecting the argument)

*(3) Aggregation.* The ArborCell prediction aggregates leaf values under these routing weights:

$$\hat{y}_c = \boldsymbol{\alpha}^\top \mathbf{v} = \mathrm{softmax}\big(\tau_2 \mathbf{K}_{(\text{tokens})} \mathbf{Q}_{(\text{CLS})}^\top\big) \mathbf{V}_{(\text{tokens})} \tag{11}$$

**Discussion:** The stated omomorphism to self-attention is positioned not as a functional replacement or a claim to superior performance, but as a conceptual bridge for theoretical interpretation. It demonstrates that the hierarchical, sparse routing of a decision tree can be reframed through the lens of a single-query attention mechanism. This perspective offers a fresh interpretative lens for understanding the fundamental inductive biases of tree-based models.

## 4 EXPERIMENTS

### 4.1 EXPERIMENTAL SETUP

**Datasets:** We evaluated ArbNN using both public benchmarks and a self-constructed industrial dataset. The public datasets, obtained from the *PyTorch Frame* project Hu et al. (2024), include Magic Gamma Telescope (MA), California Housing (CH), House 16H (H16), Jannis (JA), Diabetes130US (DI), MiniBooNE (MI), YProp 4.1 (YP), California (CA), Medical Charges (ME), and Credit (CR). We used the official splits provided by *PyTorch Frame* and randomly split the data into 80% training and 20% testing when no standard split was available. These benchmarks typically contain tens of thousands of samples but have limited representation of negative classes. Additionally, their relatively small test sets constrain the statistical reliability of model performance comparisons.

To address the limitations of existing small-scale and static academic datasets, we introduce **Tab-Credit**, a large-scale commercial credit dataset comprising roughly 1 million loan application records per month, each with 100 anonymized and carefully engineered features. TabCredit is temporally partitioned into a training set (September 2023) and an OOT test set (November 2023), enabling rigorous evaluation under temporal drift. Each record includes future risk outcomes from two to twelve months-on-book (MOB2–MOB12), with default rates that naturally evolve over time, offering a realistic benchmark for risk modeling. Unlike many curated academic datasets, TabCredit preserves its native distribution, thereby reflecting real-world challenges of scale, drift, and heterogeneity. The dataset will be released upon publication to support further research. More details about TabCredit are provided in Appendix B.

**Data preprocessing:** For GBDTs and ArbNN, we retained the native scale of each dataset, while other neural network baselines were trained on normalized inputs. For numerical features, missing values were imputed using the minimum observed value per feature, whereas categorical features were encoded using one-hot representations.

**Implementation and Training Details:** To ensure fairness in all comparative experiments, we applied consistent hyperparameter tuning procedures across all methods. In particular, tree-based baselines such as XGBoost, as well as all neural network baselines, were optimized using Optuna

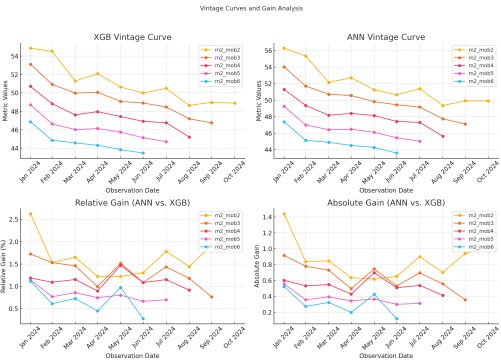

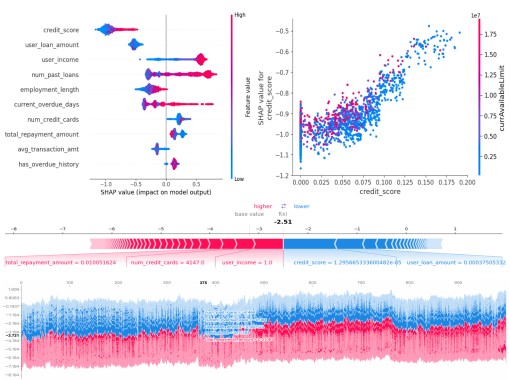

Figure 2: Vintage curves on the OOT test set of our private commercial dataset, showcasing the superior and consistently robust forecasting ability of ArbNN compared to XGBoost.

Figure 3: We use TreeSHAP to show ArbNN's tree-level interpretability on TabCredit at both the population and individual levels.

with identical search space constraints, consistent with the *PyTorch Frame* benchmark framework Hu et al. (2024). For the proposed ArbNN, model parameters were initialized using pretrained XGBoost models from the same comparison group. For classification tasks, the final prediction was computed by summing the logits produced by all tree neurons and applying a sigmoid activation. For regression tasks, the logits were directly summed, and a constant bias of 0.5 was added to improve initial convergence. We used the AdamW optimizer Loshchilov & Hutter (2017) with a learning rate of $1 \times 10^{-4}$, a batch size of 2048, and a weight decay of 0.02. The learning rate followed a two-phase schedule: a linear warmup over the first two epochs, followed by linear decay. All experiments were conducted using Python 3.8 and PyTorch on NVIDIA Tesla A100 GPUs. To emulate deterministic tree behavior while preserving differentiability, we used a fixed temperature parameter $\tau_1$ and $\tau_2$ to ensure that ArborCell units effectively approximate hard-split logic. We empirically validated this setting through ablation experiments over $\tau_1, \tau_2 \in \{5, 10, 50, 100, 200\}$ and observed that $\tau_1 = 100$ and $\tau_2 = 10$ offered the optimal trade-off between accuracy and smoothness.

**Evaluation Metrics:** To evaluate the performance of the proposed ArbNN, we used AUC-ROC for classification tasks and RMSE for regression tasks on public datasets, reflecting discriminative power and predictive accuracy, respectively. For our large-scale commercial credit-risk dataset, we conducted a vintage analysis using the Kolmogorov-Smirnov (KS) statistic on M2+@MOB2 through M2+@MOB12, where M2+ denotes delinquency beyond 31 days and MOBN represents months-on-book. We also report Lift to assess the model's ranking quality.

## 4.2 SOTA PERFORMANCE ON RESULTS

**Comparison on Public Benchmarks:** We compared ArbNN with mainstream neural models, tree models and hybrid models on the public datasets. Results are presented in Table 1. As shown, ArbNN achieves SOTA results on nearly all benchmark datasets, consistently outperforming other models, including XGBoost and LightGBM. The key to this superior performance is two-fold: effective initialization and end-to-end optimization. During initialization, ArbNN inherits the predictive power of pre-trained ensemble decision trees, positioning it at least on par with the tree models. Crucially, it then benefits from global, end-to-end fine-tuning, surpassing the local and greedy optimization typical of ensemble methods. This optimization enables it to refine split thresholds, structural weights, and leaf values holistically.

**Comparison on TabCredit:** It can be seen from Table 2 that ArbNN consistently achieves the highest KS values across all MOB points, outperforming the suboptimal baseline by margins of around 0.5% or more at each stage. This underscores ArbNN's stronger ability to discriminate between potential defaulters and non-defaulters over time, especially in a large-scale and highly heterogeneous credit-risk environment.

Beyond the KS metric, we additionally computed AUC-ROC and Lift to assess ranking quality and prioritization ability, benchmarking ArbNN against XGBoost, the strongest traditional GBDT

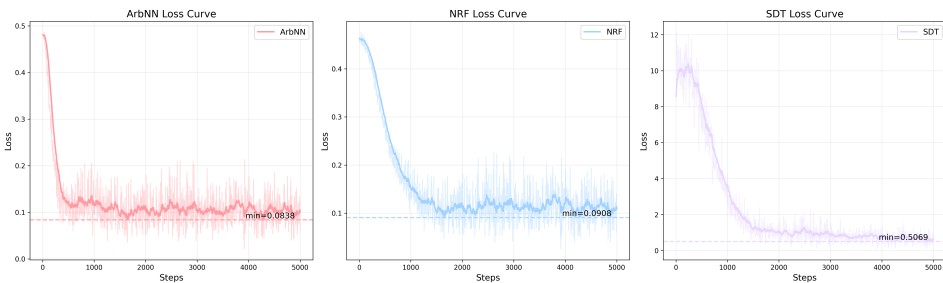

Figure 4: Training loss comparison under randomly initialized $f$ and $v$. ArbNN converges faster and to a lower loss than prior neuralized–tree models (NRF and SDT), illustrating the stability and efficiency of its one–shot multi–path routing formulation.

baseline. ArbNN achieves a slightly higher AUC-ROC (0.8194 vs. 0.8188) while also attaining a marginally better Lift score (1.8442 vs. 1.8440). These improvements, though numerically small, consistently favor ArbNN and indicate its stronger ability to prioritize high-risk borrowers, which is crucial in real-world credit modeling applications.

**Comparison on Our Private Commercial Dataset:** To showcase model performance under conditions closely resembling those of real-world production environments, we also conducted a vintage curve analysis in Figure 2 to evaluate the performance of ArbNN versus the representative model, XGBoost. The vintage curve comparison for the OOT test set demonstrates that the proposed ArbNN consistently outperforms XGBoost across all observed months from January 2024 to October 2024. While the training dataset uses December 2023 with MOB6 as the prediction label, the OOT evaluation spans MOB2 through MOB6, introducing both a 10-month time shift and variation in target definitions. Despite these temporal drifts and label heterogeneity, ANN achieves stable performance gains, with an average relative improvement of approximately 1.2% and an absolute KS gain of around 0.6. This indicates not only model robustness under long-horizon forecasting but also a stronger capacity to generalize and adapt to complex, evolving data patterns compared to XGBoost. These results reinforce the superior predictive power and temporal stability of ArbNN in real-world commercial credit risk scenarios.

### 4.3 Ablation and Interpretability Analysis

**Bidirectional Fidelity and Training Dynamics:** We first verified the bidirectional fidelity of ArbNN using the *None (Decompiled)* configuration in Table 3, which evaluates the decompiled symbolic tree without any trainable parameters. The bidirectional compilation ensures that soft neural inference and hard symbolic tree inference are matched in their ranking behavior: both KS and AUC agree up to 4–5 decimal places, indicating that interpretability and performance are aligned rather than traded off.

To further assess the optimization dynamics, we examined the $f + v$ *(Random Init)* setting, where split thresholds $f$ and leaf values $v$ are randomly initialized while the tree topology is kept fixed. Even under this uninformed initialization, ArbNN reliably converges and achieves performance that exceeds the XGBoost baseline. This demonstrates that on top of a fixed tree topology, the differentiable ArborCell formulation is able to optimize toward a strictly better solution, revealing that our continuous relaxation not only preserves the tree structure but also yields a more expressive and trainable refinement of it. The loss–curve analysis 4 further corroborates this result: with randomly initialized $f$ and $v$, ArbNN converges faster and more stably than prior neuralized–tree models such as NRF Biau et al. (2019) and SDT Frosst & Hinton (2017), highlighting the advantages of its one–shot multi–path routing and well–conditioned optimization geometry.

**Parameters Fine-Tuned Study:** All the parameters in ArborCell can be finetuned. We conducted ablation studies and analyses on the finetuned parameters based on the TabCredit dataset. Specifically, we compared the effects of fine-tuning different subsets of parameters as presented in Table 3. The configuration that fine-tunes both $f + v$ *(Decompiled Init)* achieves the best overall performance, yielding the highest KS and AUC-ROC—while preserving the original decision-path logic and model interpretability. In contrast, updating all parameters degrades performance and eliminates interpretability by destroying the tree-aligned sparse structure. Updating the tree-structure matrix

| Trainable parameters | M2+@MOB3 | M2+@MOB6 | M2+@MOB12 | AUC-ROC | Interpretability |
|---|---|---|---|---|---|
| None (Decompiled) | 53.92 | 48.19 | 39.13 | 81.88 | Preserved |
| All parameters | 53.69 | 48.09 | 39.04 | 81.82 | Lost |
| Tree-structure matrix P | 53.94 | 48.18 | 39.12 | 81.88 | Partially lost |
| Leaf values v | 54.10 | 48.38 | 39.30 | 81.91 | Preserved |
| Bias $f$ | 54.08 | 48.37 | 39.25 | 81.90 | Preserved |
| $f$ + v + P | 54.11 | 48.40 | 39.28 | 81.93 | Partially lost |
| $f$ + v (Randomly Init) | 53.95 | 48.20 | 39.17 | 81.90 | Preserved |
| $f$ + v (Decompiled Init) | 54.13 | 48.43 | 39.32 | 81.94 | Preserved |

Table 3: Study on fine-tuned parameters based on TabCredit.

alters subspace allocation and weakens the semantic alignment with decision trees, resulting in only partial interpretability.

**Axis-Aligned Decision Tree Perspective Interpretability:** To show the retained axis-aligned decision tree level interpretability of ArbNN, we first emphasized that ArborCell naturally supports a bidirectional mapping: the *Compilation* step converts a decision tree **T** into a differentiable ArborCell representation, enabling **end-to-end training**, while the *Decompilation* step maps the fine-tuned ArborCell back to a refined tree **T′**, thereby ensuring **interpretability retention**. Crucially, this allows us to apply *TreeSHAP*—the tree-specific variant of SHapley Additive exPlanations (SHAP) Lundberg et al. (2019)—which leverages the symbolic tree structure for exact and efficient feature attribution. Figure 3 provides TreeSHAP-based analyses of ArbNN's interpretability. The summary plot (top left) identifies `credit_score`, `user_loan_amount`, and `user_income` as the most influential features, consistent with domain expectations. The dependence plot (top right) further reveals meaningful feature interactions, such as between `credit_score` and `available_limit`, demonstrating ArbNN's ability to capture nuanced second-order effects. At the individual level, the force plot (center) provides per-instance explanations, while the aggregated global attribution map (bottom) highlights how positive and negative contributions are distributed across the population. Together, these visualizations confirm that ArbNN combines predictive strength with structural transparency, supporting its practical deployment in high-stakes domains such as credit risk prediction.

## 5 CONCLUSION AND DISCUSSION

The proposed ArbNN bridges symbolic decision trees and differentiable neural models through the ArborCell module, enabling globally interpretable prediction with end-to-end optimization. By leveraging tree initialization, ArbNN inherits a strong inductive bias while being fine-tuned through gradient descent. It achieves state-of-the-art performance on public tabular benchmarks and has been deployed in a large-scale credit-risk system, where it is handling millions of applications and billions of dollars in decisions. We also released the TabCredit dataset to facilitate realistic evaluation.

Despite these advantages, ArbNN has limitations. First, it does not learn tree structure entirely from scratch: routing is inherited from a decision-tree prior, and only split thresholds and leaf values are optimized end-to-end. Developing fully differentiable structure learning remains an important direction for future work. Second, the framework was tailored to tabular domains; extending ArbNN to multimodal settings such as text, images, and sequential data remains open. Additional progress in scalable routing, streaming training, and neural–symbolic model design would further enhance its applicability.

REPRODUCIBILITY STATEMENT

We have made several efforts to ensure reproducibility. The proposed ArbNN model is fully described in Section 3, with its mathematical equivalence to self-attention formally proved. Details of datasets, preprocessing, and feature characteristics are provided in Section 4 and Appendix B. Experimental settings, including hyperparameter search procedures, training configurations, and evaluation metrics, are described in Section 4.1. The tree parsing algorithm for converting GBDTs into ArborCells is included in Appendix A. To further support reproducibility, both the ArbNN implementation and the TabCredit dataset will be released upon acceptance.

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

# Appendices

## A  Tree Parsing Algorithm

To bridge gradient-boosted decision trees (GBDTs) and the proposed Arboreal Neural Network (ANN), we designed a tree parsing algorithm that systematically converts each tree in a GBDT ensemble into a differentiable ArborCell representation. The procedure ensures that the original inductive bias and structure of the tree are faithfully preserved, while making the resulting model fully trainable under end-to-end optimization. The algorithm consists of the following key steps:

---

**Algorithm 1:** Tree Parsing Algorithm

---

**Input:** Decision tree $T$; input dimension $t$

**Output:** Number of inner nodes $n_i$, leaf nodes $n_j$; structure matrix $\mathbf{P}$, weight matrix $\mathbf{W}$, splits vector $f$,
      leaf values $\mathbf{v}$, leaf node set $\mathcal{L}$

**Initialize:** leaf values $\mathbf{v}$, leaf node set $\mathcal{L}$, internal node set $\mathcal{I}$, split features $s$, split thresholds $f$, path record
  $\mathcal{P}$, direction record $\mathcal{D}$; counters $l \leftarrow 0$, $i \leftarrow 0$

**Define recursive function DFS**$(n, \mathcal{P}, \mathcal{D})$: **begin**
    **if** $n$ *is a leaf* **then**
        **if** $n.id \notin \mathcal{L}$ **then**
            $\mathbf{v}[n.id] \leftarrow n.\text{leaf\_value}$; $\mathcal{L}[n.id] \leftarrow l$; $l{+}{+}$
        $\mathcal{P}[n.id] \leftarrow \mathcal{P} + [n.id]$; $\mathcal{D}[n.id] \leftarrow \mathcal{D}$
    **else**
        **if** $n.id \notin \mathcal{I}$ **then**
            $\mathcal{I}[n.id] \leftarrow i$; $s[n.id] \leftarrow n.\text{feature\_index}$; $f[n.id] \leftarrow n.\text{split\_threshold}$; $i{+}{+}$
        **foreach** *child $n'$ in $n$.children* **do**
            $\mathcal{D}' \leftarrow \mathcal{D} + [\delta(n', n)]$; **DFS**$(n', \mathcal{P} + [n.id], \mathcal{D}')$

**Function** $\delta(n', n) = \begin{cases} 1 & \text{if } n'.id = n.\text{right\_child} \\ -1 & \text{otherwise} \end{cases}$

**Main Procedure**: **begin**
    **DFS**$(T, [\,], [\,])$; $n_i \leftarrow |\mathcal{I}|$, $n_j \leftarrow |\mathcal{L}|$
    Initialize $\mathbf{W} \in \mathbb{R}^{n_i \times t}$, $f \in \mathbb{R}^{n_i}$ with zeros
    **foreach** $(n.id, idx) \in \mathcal{I}$ **do**
        $\mathbf{W}_{idx, s[n.id]} \leftarrow 1$; $f_{idx} \leftarrow c[n.id]$
    Initialize $\mathbf{P} \in \mathbb{R}^{n_j \times n_i}$ with zeros
    **foreach** $(n.id, l') \in \mathcal{L}$ **do**
        $p \leftarrow \mathcal{P}[n.id]$; $d \leftarrow \mathcal{D}[n.id]$
        **foreach** $(nid, d_i) \in (p, d)$ **do**
            **if** $nid \in \mathcal{I}$ **then**
                $\mathbf{P}_{l', \mathcal{I}[nid]} \leftarrow d_i$
    $\mathbf{P} \leftarrow$ **ApplyWeights**$(\mathbf{P})$; reorder $\mathbf{v}$ based on $\mathcal{L}$
    **return** $n_i$, $n_j$, $\mathbf{P}$, $\mathbf{W}$, $f$, $\mathbf{v}$, $\mathcal{L}$

**Function ApplyWeights**$(\mathbf{P})$: **begin**
    $k \leftarrow$ max number of non-zeros per row in $\mathbf{P}$; $w_j \leftarrow (1/2)^j$ for $j = 0, \ldots, k-1$
    **foreach** *row $i$ in $\mathbf{P}$* **do**
        $I_{nz} \leftarrow$ indices of non-zero elements in $\mathbf{P}_i$
        **foreach** $(j, idx) \in$ *enumerate*$(I_{nz})$ **do**
            $\mathbf{P}_{i, idx} \leftarrow \mathbf{P}_{i, idx} \cdot w_j$
    **return** $\mathbf{P}$

---

# B  TABCREDIT DATASET

To overcome the limitations of existing small-scale academic benchmarks, we introduce **TabCredit**, a proprietary commercial credit dataset covering a two-month period, comprising approximately **1 million unique user loan application records per month**, totaling around **2 million records**. Each record has 100 anonymized and extensively engineered features.

TabCredit is partitioned into two parts: a *training set* consisting of users with successful loan applications from September 2023, and an *out-of-time (OOT) test set* containing users from November 2023. This temporal split enables strict in-time and OOT evaluation, ensuring rigorous assessment of model generalization. For each record, TabCredit provides future risk outcomes for both training and OOT sets, including monthly delinquency indicators from two to twelve months-on-book (MOB2–MOB12). Here, **MOB** refers to the number of months after loan origination, and delinquency is measured using the industry-standard **M2+** definition, which denotes accounts with payments overdue by *31 days or more*. Accordingly, the reported rates correspond to the cumulative proportion of users who have ever entered the M2+ state by each MOB horizon.

Table 4 illustrates representative cumulative M2+ delinquency rates across two cohorts. As expected, delinquency starts at relatively low levels (around 0.2–0.3% at MOB2), but grows monotonically with MOB, reaching approximately 5–6% by MOB12. This steady increase reflects the temporal dynamics of credit portfolios and provides a realistic setting for developing and evaluating risk forecasting models.

| Cohort | MOB2 | MOB3 | MOB4 | MOB5 | MOB6 | MOB7 | MOB8 | MOB9 | MOB10 | MOB11 | MOB12 |
|---|---|---|---|---|---|---|---|---|---|---|---|
| 202309 | 0.26% | 0.67% | 1.35% | 2.05% | 2.66% | 3.40% | 4.04% | 4.62% | 5.16% | 5.64% | 6.16% |
| 202311 | 0.20% | 0.67% | 1.26% | 1.82% | 2.57% | 3.22% | 3.80% | 4.33% | 4.86% | 5.40% | 5.92% |

Table 4: Example cumulative **M2+** delinquency rates (defined as 31+ days past due) across different months-on-book (MOB2–MOB12) for two representative cohorts in TabCredit. Delinquency accumulates steadily with MOB, highlighting the temporal progression of credit risk.

In addition to overall dataset statistics, we provide representative examples of feature characteristics from TabCredit in Table 5. The dataset contains heterogeneous attributes drawn from both credit bureau records and behavioral signals. For instance, **Credit Bureau Main Cardholder Score** (99% coverage) captures a user's repayment history, while **Credit Bureau Loan Record Last Update Date** (99%) reflects the freshness of a borrower's bureau file. Utilization-related indicators, such as the **Max Credit Utilization Ratio (Past 3 Months, 100% coverage)** and the **Current Fixed Credit Limit (100%)**, characterize users' recent borrowing behavior and credit capacity. Finally, the **Number of Inquiry Institutions in the Past 6 Months** (coverage $\sim$87%) measures the intensity of recent credit-seeking behavior, often serving as an early warning indicator of potential default risk. Together, these features illustrate the diversity, realism, and predictive richness of TabCredit.

| Feature | Type | Coverage | Range / Example Values | Description |
|---|---|---|---|---|
| Credit Bureau Main Cardholder Score | Numerical (ordinal) | 99% | 300–900 (e.g., 650, 720) | Bureau-based credit score for existing cardholders |
| Credit Bureau Loan Record Last Update Date | Date / Timestamp | 99% | e.g., 2023-07-15 | Last update date of loan record in credit bureau |
| Max Credit Utilization Ratio (Past 3 Months) | Numerical (ratio) | 100% | 0–1 (e.g., 0.25, 0.83) | Maximum utilization ratio of credit line in past 3 months |
| Current Fixed Credit Limit | Numerical (continuous) | 100% | 0–300k (e.g., 50,000; 120,000) | Current fixed credit line granted to the user |
| Number of Inquiry Institutions in the Past 6 Months | Numerical (count) | 87% | 0–15 (e.g., 2, 6) | Count of distinct institutions inquiring about user's credit in the past 6 months |

Table 5: Representative feature characteristics from the TabCredit dataset. Coverage denotes the proportion of users with non-missing values for each feature. Notably, frequent inquiries (last row) often serve as strong risk indicators in real-world credit scoring.

In summary, TabCredit combines scale, feature richness, temporal partitioning, and longitudinal outcomes. It provides a realistic and challenging benchmark for evaluating not only predictive accuracy, but also interpretability and stability of tabular learning methods in credit risk modeling.

