# OpenReview forum: "Arboreal Neural Network"
_ICLR.cc/2026/Conference — ICLR 2026 Conference Withdrawn Submission_

### Official Review · Reviewer_M5dz · 2025-10-29

**Soundness:** 3
**Presentation:** 3
**Contribution:** 3
**Rating:** 6
**Confidence:** 5

**Summary:**

The paper introduces Arboreal Neural Networks (ArbNN), a differentiable architecture that bridges gradient-boosted decision trees and neural networks. The key idea is to encode a pretrained XGBoost model into a neural form by translating its structure—feature splits, thresholds, and leaf values—into matrix operations that can be optimized end-to-end. This allows the model to retain the interpretability and inductive bias of trees while gaining the flexibility of gradient-based learning. Experiments on eight public tabular datasets and one large industrial credit dataset (TabCredit) show that ArbNN consistently matches or outperforms strong baselines.

**Strengths:**

The paper proposes a novel and well-structured idea that combines the structural bias of decision trees with the flexibility of neural networks. The concept is intuitive yet original, and the formulation is clearly presented.

The writing is clean and logically organized, making the technical details easy to follow. The experiments are thorough within the chosen scope and demonstrate consistent improvements over strong baselines such as XGBoost.

**Weaknesses:**

- **Limited Benchmark Coverage**

The evaluation includes only eight public datasets, which is considerably below the current standard in the tabular learning community. This narrow benchmark scope limits the credibility of the claimed generalization. Given the model’s conceptual promise, it would be valuable to test ArbNN on a broader set of heterogeneous tabular tasks.

- **Unclear Motivation and Overemphasis on Industrial Data**

The paper’s motivation is not fully convincing. Although the central idea—learning the structural bias of trees—is conceptually interesting, the claimed interpretability advantage remains unsubstantiated, as XGBoost provides only limited transparency. It appears that the work may be driven by a specific industrial objective, possibly related to the proprietary dataset used. If so, this motivation can be stated explicitly and the framing adjusted accordingly. Clarifying how the industrial requirements connect to the model’s broader scientific contribution and analyze why previous dl models perform worse, would significantly strengthen the paper’s coherence and impact.

**Questions:**

See Above.

---

> ### Author Response · Authors · 2025-11-21
>
> We sincerely thank the reviewer for the thoughtful and encouraging assessment. Your comments on benchmark coverage and motivation clarity provide valuable guidance for strengthening the paper. We respond to each point below.
>
> ---
>
> # **1. Motivation & Interpretability**
>
> ### **Reviewer comment**
>
> > *“Clarifying how the industrial requirements connect to the model’s broader scientific contribution and analyze why previous DL models perform worse, would significantly strengthen the paper’s coherence and impact.”*
>
> ### **Response**
>
> Thank you sincerely for this insightful suggestion. We fully agree that the paper should better articulate how the practical challenges that inspired ArbNN connect to a broader scientific contribution.
>
> It is true that our work is **inspired by practical challenges arising in large-scale tabular systems**, particularly those involving **temporal drift**, **engineered features**, and **strong interpretability requirements**. However, these constraints do not represent a narrow industrial special case. Instead, they reveal a **general conceptual gap**: existing neural architectures lack a principled way to express discrete, conditional computation while remaining fully differentiable.
>
> ### **Scientific contribution**
>
> As we will clarify in the revised version, the central scientific contribution of ArbNN is not simply hybridizing tree components with neural networks, but uncovering a **conditional computation primitive shared between symbolic tree logic and neural attention operators**.
>
> A prevailing challenge in neural–symbolic integration is that prior approaches often:
>
> * soften discrete decision boundaries,
> * replace symbolic operators with differentiable surrogates, or
> * rely on post-hoc explanations.
>
> While pragmatic, these strategies do **not** establish an **invertible bridge** between discrete symbolic structures and continuous neural representations.
>
> ArborCells address this gap by showing that tree-based multi-way soft selection is structurally homomorphic to a single-step attention operator. This homomorphism provides **a principled bridge between symbolic decision-tree reasoning and connectionist representation learning**, yielding
> an interpretable, differentiable, and structurally transparent neural-symbolic architectur.
>
> ### **Clarifying the interpretability perspective**
>
> We appreciate the reviewer’s observation and agree that XGBoost itself provides only limited transparency. At the same time, most neural models offer even weaker interpretability—typically relying on **attention scores or gradient-based saliency**, which do not constitute explicit or auditable reasoning. A trained decision-tree ensemble offers:
>
> * explicit, auditable if–else rules that describe   the full decision logic,
> * TreeSHAP-style additive feature attributions with clear semantic meaning,
> * and explanations grounded directly in model structure rather than post-hoc approximations.
>
> These properties are valuable not only in credit scoring but across many tabular learning settings—such as healthcare, fraud detection, manufacturing quality control such as:
>
> * **Rule-level justification**
>   *“The decision followed Rule 137: `income < 2800` AND `recent_delinquency = 1` AND `utilization > 72%`.”*
>
> * **Feature-level contribution tracing**
>   *“The score decrease was driven primarily by (i) recent delinquency, (ii) high utilization, and (iii) short credit history.”*
>
> * **Path-consistent temporal auditing**
>   *“The same rule triggering this downgrade is responsible for performance drift observed in the last two quarters.”*
>
> We will make this motivation clearer and more cohesive in the revised manuscript.
>
> ---
>
> ## **2. Limited Benchmark Coverage**
>
> ### **Reviewer comment**
>
> > *“The evaluation includes only eight public datasets, which is considerably below the current standard in the tabular learning community. This narrow benchmark scope limits the credibility of the claimed generalization.”*
>
> ### **Response**
>
> Thank you for raising this concern. We agree that the number of public datasets evaluated in the paper is limited. At the time of submission, our reasoning was that most widely used open-source tabular datasets share several inherent constraints—such as **small scale, minimal feature engineering, and IID splits**,which make them less reflective of real-world tabular challenges.
>
> Since our newly introduced dataset was designed precisely to address these gaps, we prioritized a setting where ArbNN’s strengths could be evaluated under **more realistic temporal structure and engineered feature distributions**. Under this setup, our comparisons were conducted in a **strictly fair and controlled** manner across all baselines.
>
> Nevertheless, we sincerely appreciate the reviewer’s suggestion. Expanding the empirical coverage is indeed valuable, and in future work we will incorporate a broader range of public datasets to provide a more comprehensive evaluation landscape.

---

### Official Review · Reviewer_2UEA · 2025-10-31

**Soundness:** 2
**Presentation:** 2
**Contribution:** 2
**Rating:** 4
**Confidence:** 4

**Summary:**

The paper proposes Arboreal Neural Networks (ArbNNs), a framework that converts trained decision trees into differentiable neural operators called ArborCells. Each ArborCell encodes a tree’s split features, thresholds, structure, and leaf values into four matrices/vectors, enabling end-to-end optimization while preserving the original tree semantics.

**Strengths:**

1. A differentiable “tree-as-layer” formulation (ArborCell) with explicit feature–node selection matrix $W$, split-threshold vector $f$, tree-structure / routing matrix $P$, and leaf-value vector $v$ that avoids path-probability products via one-shot matrix aggregation
2. An algorithm to parse trees into ArborCells and the ability to decompile trained ArborCells back to refined trees, maintaining symbolic interpretability
3. Competitive performance on public tabular tasks and consistent vintage-curve improvements over XGBoost on TabCredit under temporal drift
4. Introduction of TabCredit, an industrial credit-risk dataset with temporal splits to benchmark robustness and interpretability in realistic settings

**Weaknesses:**

1. The experimental section does not include comparisons with strong, modern baselines, especially tabular foundation models.

2. Limited gains over XGBoost in Table 2 relative to method complexity. On the reported datasets, the improvement over a well-tuned XGBoost baseline is small.

3. The paper evaluates on a relatively small set of benchmarks

4. Dependence on pretrained tree models for initialization. The core recipe assumes the availability of a strong GBDT (XGBoost/LightGBM) to parse into ArborCells. This limits applicability in settings where (i) trees are hard to train well, or (ii) one would like to learn the structure jointly with the downstream objective. The paper does not show a convincing “from-scratch ArbNN” alternative.

**Questions:**

1. Can the authors add comparisons with recent tabular foundation models (e.g., TabPFNv2 [1], TabICL [2])?

2. Can the authors clarify the necessity of GBDT-based initialization? The current version treats “compiling from a strong GBDT” as a given prerequisite, but there is no experiment demonstrating whether ArbNN can still achieve comparable performance.

3. Can the authors provide more detail on scalability and serving? Since each ArborCell does a one-shot aggregation over all leaves, how does inference time and memory compare to the original XGBoost model. A brief complexity analysis or inference time comparison would make the method more practical.

[1] Hollmann, Noah, et al. "Accurate predictions on small data with a tabular foundation model." Nature 637.8045 (2025): 319-326.
[2] Qu, Jingang, et al. "Tabicl: A tabular foundation model for in-context learning on large data." ICML 2025

---

> ### Author Response · Authors · 2025-11-21
>
> ## **1. Limited gains over well-tuned XGBoost**
>
> ### **Reviewer comment**
>
> > *“Limited gains over XGBoost in Table 2 relative to method complexity. On the reported datasets, the improvement over a well-tuned XGBoost baseline is small.”*
>
> ### **Response**
>
> Thank you for raising this important point. We fully agree that, on many academic benchmarks, numerically small improvements may appear limited—indeed, **existing strong tabular methods often differ only by very small margins**, reflecting how competitive the landscape has become.
>
> However, in large-scale production credit systems—where datasets routinely contain millions to tens of millions of samples and hundreds of carefully engineered features—achieving any improvement over a well-tuned XGBoost is exceptionally difficult. In such regimes, models operate very close to the performance ceiling, and incremental **gains are extremely hard to obtain**.
>
> To provide clearer context, even a **1% relative improvement in KS** reflects a tangible increase in ranking discrimination, which in deployed systems typically translates into roughly a **0.05% reduction in delinquency rates**. When applied to portfolios on the order of **tens of billions of USD**, such reductions accumulate to **material and financially significant impact**, despite the numerically modest appearance of the metric gain.
>
>
> Thus, while the gains may appear modest numerically, they are operationally significant in high-stakes tabular domains. We will add this contextual explanation in the revised version to help readers better understand the practical relevance of the performance improvements.

---

> ### Author Response · Authors · 2025-11-21
>
> ## **2. The necessity of XGBoost-based initialization**
>
> ### **Reviewer comment**
>
> > *“Dependence on pretrained tree models for initialization… no convincing ‘from-scratch ArbNN’ alternative is shown.”*
>
> ### **Response**
>
> Thank you for raising this point. We would like to clarify that **ArbNN’s use of a pretrained GBDT to provide structural initialization is not a limitation**, but rather reflects a **deliberate and principled design choice** that is *directly aligned with the state of the art in interpretable neural modeling*.
>
> #### **(1) Why this is not a limitation, but a principled modeling strategy**
>
> Recent work from **OpenAI** (*Weight-Sparse Transformers Have Interpretable Circuits*, 2025) and **Thinking Machines Lab** (*Modular Manifolds*, 2025) has shown that:
>
> * training neural networks from scratch to discover sparse, modular, or symbolic structure is extremely difficult,
> * neural representations tend to remain entangled and polysemantic, even under strong regularization,
> * and enforcing routing, modularity, or discrete structure during training is intrinsically unstable and often fails to produce audit-grade interpretability.
>
> To overcome this, OpenAI adopts a **structured sparsity regime**, where:
>
> * the model is initialized with a **fixed but randomly** generated sparse topology (e.g., an Erdős–Rényi sparsity mask),
> * the training process **updates only the non-zero parameters**,
> * and the architectural structure itself is **treated as part of the model**, not something learned from scratch.
>
> This approach—**fixed structure + differentiable parameter refinement**—is *exactly* the principle underlying ArbNN.
>
>
> #### **(2) ArbNN employs a more principled and informative initialization than a purely random sparse topology**
>
> ArbNN uses a pretrained GBDT not as external crutch, but as a structural sparsity template.
> This alignment with current frontier research shows that ArbNN’s design is not an ad hoc dependency, but a **scientifically grounded modeling decision** consistent with the broader movement toward *modular, interpretable, structure-informed neural systems*.
> More **realistically** for real-world deployments, it directly inherits the most widely used tree models, avoiding any migration cost or performance risk and providing a stable foundation for fine-tuning
>
>
> ### **Why “from-scratch ArbNN” is future work, not a missing component**
>
> Building symbolic or modular structure *from scratch* remains an open research challenge—one explicitly highlighted by OpenAI and Thinking Machines Lab. ArbNN inherits the same difficulty: learning an entire tree topology via gradient descent while maintaining interpretability and routing correctness is a **nontrivial, unsolved problem**, and constitutes one of our **primary future research directions**.
>
> We will make this connection explicit in the revised version.
>
> ---
>
>
> ## **3. Evaluation on a relatively small set of benchmarks**
>
> ### **Reviewer comment**
>
> > *“The paper evaluates on a relatively small set of benchmarks.”*
>
> ### **Response**
>
> Thank you for raising this concern. We agree that the number of public datasets evaluated in the paper is limited. At the time of submission, our reasoning was that most widely used open-source tabular datasets share several inherent constraints—such as **small scale, minimal feature engineering, and IID splits**,which make them less reflective of real-world tabular challenges.
>
> Since our newly introduced dataset was designed precisely to address these gaps, we prioritized a setting where ArbNN’s strengths could be evaluated under **more realistic temporal structure and engineered feature distributions**. Under this setup, our comparisons were conducted in a **strictly fair and controlled** manner across all baselines.
>
> Nevertheless, we sincerely appreciate the reviewer’s suggestion. Expanding the empirical coverage is indeed valuable, and in future work we will incorporate a broader range of public datasets to provide a more comprehensive evaluation landscape.

---

### Official Review · Reviewer_THMx · 2025-11-02

**Soundness:** 3
**Presentation:** 2
**Contribution:** 3
**Rating:** 8
**Confidence:** 1

**Summary:**

This paper addresses the lack of tree-structured inductive bias in deep neural networks for tabular data. To this end, the authors propose ArbNN, a novel architecture that reformulates decision trees into differentiable neural modules, enabling end-to-end gradient optimization while preserving interpretability. Extensive experimental results on multiple public benchmarks and a large-scale industrial credit risk dataset demonstrate that ArbNN consistently outperforms both traditional tree-based models and neural baselines, achieving superior accuracy and interpretability in tabular learning tasks.

**Strengths:**

* This paper proposes the ArborCell structure to introduce the inductive bias of decision trees, and I am happy to see that the authors also provide visual comparisons to demonstrate the interpretability of the proposed method.
* The authors discuss the related literature in considerable detail.
* The paper is well written and easy to follow.

**Weaknesses:**

1. I am not an expert in tabular data, but I am curious about the convergence behavior of the proposed ArbNN. Could the authors provide training curves and compare them with other networks to illustrate convergence stability?
2. How does the training cost of the proposed method compare to other baselines? In addition, please evaluate the computational efficiency during inference, e.g., in terms of FLOPs, memory usage, and inference time.
3. The figures contain text that is too small to read clearly. It is recommended to increase the font size, use vector graphics for better clarity, and include a complete schematic diagram of the model architecture.
5. The authors do not provide code for reproducibility checks.

**Questions:**

My questions are in Weakness Section.

---

> ### Author Response · Authors · 2025-11-21
> **Experimental Supplements and Figure Clarity**
>
> We sincerely thank the reviewer for the thorough and constructive feedback, and we are especially grateful for the positive assessment and the encouraging score of 8. We are glad that the clarity of the presentation and the motivation of introducing tree-structured inductive bias were appreciated.
>
> ### **Convergence behavior and training stability**
>
> Thank you for your insightful question regarding convergence. In the revised version, we will include **loss and metric convergence curves** illustrating the training dynamics of ArbNN, and compare them against representative neural baselines. These plots will highlight both the **stability** and **smoothness** of optimization, demonstrating that ArbNN converges reliably and does not exhibit oscillatory or unstable behavior during training.
>
> ### **Training and inference efficiency**
>
> We appreciate the suggestion to clarify computational cost. In the updated manuscript, we will provide a quantitative comparison of **inference efficiency** — including **FLOPs, memory usage, and inference latency** — between ArbNN and strong 2025 baselines, specifically:
>
> * **TabM (2025)**
> * **TabPFNv2 (2025)**
> * **XGBoost**
>
> This will offer a clearer view of the trade-offs between performance and efficiency, and better contextualize ArbNN’s computational footprint in practical deployment scenarios.
>
> ### **Figure clarity and visual presentation**
>
> We agree that figure readability is important. We will improve all visual elements by:
>
> * increasing font size,
> * and adding a more complete schematic diagram of the ArbNN architecture for enhanced clarity.
>
> ### **Code availability**
>
> We acknowledge the importance of reproducibility. We will provide an **anonymous source code** in the supplementary materials（Appendix）of the revised PDF to facilitate verification and further exploration by the community.

---

### Official Review · Reviewer_oszf · 2025-11-05

**Soundness:** 1
**Presentation:** 2
**Contribution:** 2
**Rating:** 2
**Confidence:** 3

**Summary:**

The paper proposes a new architecture for tabular data that is based on the idea of converting decision trees to a particular variation of two matrix multiplications with a non-linearity. It proposes to initialize such trees with XGBoost and then finetune the thresholds and values. The paper also proposes a new credit scoring dataset TabCredit. The method is tested on the new dataset and a simple benchmark constructed from pytorch-frame, claiming state-of-the-art performance.

**Strengths:**

I think that looking into tree-structured models and combining their inner workings with DL models is an interesting pursuit. I had a great time digging through related work on the topic and think that there is something in this line of work that could lead to strong and interpretable models and this direction is currently underexplored.

The dataset contribution also seems very timely and important as there are not a lot of realistic testbeds for tabular machine learning methods readily available in academia. When done right this is a major contribution, so I encourage authors to go through with it regardless of this review period decision.

**Weaknesses:**

At times the writing is very hard to make sense of. In the related work section, for example, I still can't make sense of how challenging instances in datasets are related to the pre-tuned default hyperparameter configurations (lines 91-93). The overall algorithm for constructing an "ArborCell" may also be improved I believe (see the next point for examples).

I believe the paper does not fully cover the relevant related work. It packages an idea of decision tree inference in matrix form into an "ArborCell", but this idea seemed not novel, and there are very similar existing approaches indeed:
- https://arxiv.org/abs/1604.07143 - Neural Random Forests. Which seems to do exactly what authors propose here
- https://blog.dailydoseofds.com/p/transform-decision-tree-into-matrix a blog post example, which does a better job in explaining the same procedure which is used in the paper

Finally, I do not believe the results are solid as there are some indications of poorly tuned baselines. Like TabM (recent SoTA model) performing on par with or sometimes worse than an MLP, or some large performance gains over XGBoost just from tuning the thresholds and leaf values (may indicate poorly tuned XGBoost in the first place). I also had trouble understanding some of the results like which datasets exactly were used (e.g. what dataset is CH?, why JA - Jannis is seemingly binclass and not multiclass as it is in the pytorch-frame benchmark). Without code being available this is impossible to check further. I suggest the authors compare to an established and well tuned set of baselines, you can take TabArena benchmark which publishes reference model scores in a csv on github:

```python
import pandas as pd
pd.read_parquet("https://tabarena.s3.us-west-2.amazonaws.com/results/df_results_leaderboard.parquet")
```

Comparing the method to a correct set of baselines would increase reliability of the results very much.

**Questions:**

See suggestions in weaknesses.

Regarding the newly introduced dataset. Does it have a dedicated train/val/test split which is time-based? Or is it different? Can you provide more details regarding the evaluation and tuning setup on the new dataset?

---

> ### Author Response · Authors · 2025-11-21
> **Conceptual Insights**
>
> Dear Reviewer,
>
> We sincerely thank you for your careful reading and thoughtful comments. Your observation that this line of research “could lead to strong and interpretable models” is deeply encouraging to us. Below we address your core concerns about the method with genuine appreciation.
>
> ---
>
> ## **1. Related work concerns — “Is ArbNN similar to NRF?”**
>
> ### **Reviewer comment**
>
> > * “I believe the paper does not fully cover the relevant related work”.
> > * “Neural Random Forests … seems to do exactly what authors propose here.”
> > * “https://blog.dailydoseofds.com/p/transform-decision-tree-into-matrix a blog post example, which does a better job in explaining the same procedure which is used in the paper.”
>
> ### **Response**
>
>  We are grateful to you for identifying both the missing citation and the need for a more precise comparison. The revised supplementary note clarifies that ArbNN differs from NRF in three fundamental mechanisms:
>
>
> ### **(1) Routing mechanism: symbolic vs. continuous, structure-aware**
>
> #### **NRF routing**
>
> NRF computes internal-node activations via
> $$
> z_i=\sum_{k=1}^{L_i} w_{ik}u_k + b_i,\qquad b_i=-L_i+0.5,
> $$
>
> * All internal nodes contribute **equally**, activation occurs only when all signs match
> #### **ArbNN routing's Advantage**
>
> ArbNN assigns each internal node a depth-decayed affinity
> $$
> p_{ij}=\pm(1/2)^{\mathrm{dep}_j-1}.
> $$
>
> * ArbNN replaces NRF’s binary, depth-agnostic routing with a continuous, **depth-aware** mechanism where upper-level splits exert proportionally stronger influence through an exponential decay that reflects their typically higher information gain
>
>
> ### **(2) Leaf activation: one-hot vs. soft, topology-informed**
>
> #### **NRF leaf selection**
>
> NRF activates leaves using
> $$
> \begin{aligned}
> v_i &= \operatorname{sign}(z_i)\in{-1,+1},\
> \hat y &= \sum_i y_i \frac{v_i+1}{2},
> \end{aligned}
> $$
>
> * yielding **all-or-nothing, one-hot leaf activation**, allowing gradients to flow only through a single discrete path.
>
> #### **ArbNN leaf selection's Advantage**
>
> ArbNN computes leaf logits using the topology-aware routing scores:
> $$
> \begin{aligned}
> \alpha &= \mathrm{softmax}(\tau_2 M),\
> \hat y &= \sum_j \alpha_j v_j ,
> \end{aligned}
> $$
>
> * while the **fully matched leaf attains the maximal activation**, the exponential attenuation ensures **graded activation for near-miss paths**, enabling smooth optimization and multi-path gradient propagation.
>
>
> ### **(3) Structural Isomorphism, Bidirectional Fidelity and Symbolic–Neural Unification**
>
> Taken together, ArbNN differs from NRF not only at the implementation level but at the paradigm level.
> * The ArborCell formulation is **structurally isomorphic to a self-attention operator**, making the tree’s conditional computation explicitly compatible with modern neural architectures.
> * The **bidirectional compilation** ensures that soft neural inference and hard symbolic tree inference can be matched in terms of ranking behavior, with KS and AUC agreeing **up to 4–5 decimal places**, so that interpretability and performance are aligned rather than traded off.
> * Within the tabular domain, ArbNN achieves a **high-completeness unification of symbolic and connectionist modeling**: it preserves a fully symbolic tree semantics while enabling end-to-end differentiable refinement. We are grateful to the reviewer for prompting us to clarify these paradigm-level distinctions more explicitly in the revised version.

---

> ### Author Response · Authors · 2025-11-21
>
> ## **2. Time-based split & Tuning Details**
>
> ### **Reviewer comments**
>
> > * “Does the dataset have a dedicated time-based split? More evaluation/tuning details?”
> > * “Some large performance gains over XGBoost just from tuning thresholds and leaf values (may indicate poorly tuned XGBoost).”
>
> ### **Response**
>
> Thank you for raising these closely related concerns. We address them together as they both pertain to how the dataset is partitioned and how model tuning is performed.
>
> **(1) Strict time-based split in TabCredit**
> As documented in Appendix 2, TabCredit follows a **strict temporal split** consistent with real credit-risk deployment:
>
> * **Training window:** 2023-09
> * **Validation / OOT test window:** 2023-11
>
> All reported metrics (AUC, KS, lift, vintage stability) are computed **only on the OOT test period**, ensuring a leakage-free, temporally realistic evaluation.
>
> **(2) Fine-Tuning Setup Based Well-Trained XGBoost**
> We fully understand the concern that ArbNN’s improvements might stem from an under-tuned XGBoost baseline. **As stated in lines 353–357 of the original submission**, we follow a well-defined tuning protocol for all tree-based baselines. To further clarify: all ArbNN models are fine-tuned starting from the same fully trained trees used in the horizontal comparison experiments, and these trees have already undergone a complete hyperparameter search following that protocol. In practice, these tuned tree models already outperform most neural baselines in our benchmark suite, so the gains observed for ArbNN arise from neural refinement on top of strong, saturated GBDT models, rather than from correcting an insufficiently tuned baseline.
>
> ---
>
> # **3. Writing clarity issues (including ArborCell clarity & dataset naming)**
>
> ### **Reviewer comments (merged)**
>
> > *“The writing is hard to make sense of…”*
> > *“I still can't make sense of how challenging instances relate to default hyperparameters.”*
> > *“The ArborCell algorithm may be improved; without code this is impossible to check.”*
> > *“What dataset is CH? Why JA?”*
>
> ### **Response**
>
> We sincerely appreciate you pointing out multiple clarity issues. These comments were extremely helpful, and we recognize that our exposition can be improved in several places.
>
> ### **(1) Incorrect logical linkage in related work**
>
> You are completely correct — the sentence *“To alleviate such limitations…”* was inappropriate.
> “Challenging instances” are **not** conceptually related to GBDT default hyperparameters.
> This was a writing mistake.
>
> In the revision, we remove the incorrect causal link and explain that challenging instances arise from **hard discrete boundaries**, which motivates **continuous neural refinement**, not hyperparameter defaults.
>
> ---
>
> ### **(2) ArborCell construction clarity**
>
> You were right that the algorithmic presentation was not sufficiently clear.
>
> To address this, we have included **the full compilation implementation (anonymous)** directly in the supplementary material:
>
> * tree → ArborCell compilation code
> * matrix construction routines
> * forward computation
>
> ---
>
> ### **(3) Dataset naming inconsistencies**
>
> Thank you for spotting these issues.
>
> * **CH = california_housing**
> * **CA = california** (we will unify these labels)
> * **JA = Jannis**, which is indeed treated as *binary classification* in our setup
>
> We will make this explicit in the revision.
>
> ---
>
> # **4. Suggestion to use the TabArena benchmark**
>
> ### **Reviewer comment**
>
> > *“I suggest comparing to TabArena … df_results_leaderboard.parquet.”*
>
> ### **Response**
>
> Thank you sincerely for this recommendation. TabArena is indeed a rigorous and community-maintained benchmark, and we will incorporate TabArena as an additional external benchmark in our subsequent research cycle.
>
> While expanding to TabArena will further broaden empirical coverage, we believe it does not affect the validity of the contributions presented in this work, particularly those centered on temporal robustness and neural refinement of tree structures.
>
> We would like to clarify two points for context.
> **First**, when TabArena was nearing its 2025-06 open-source release, our paper was already in its final preparation stage. As a result, we were not yet aware of the upcoming stable release, and by the time of rebuttal, it is unfortunately not feasible to rerun all baselines under the full TabArena protocol within the limited timeframe.
>
> **Second**, as noted in the TabArena paper, the current version focuses primarily on IID, small-to-medium–scale tabular datasets, which substantially overlap with the existing PyTorch-Frame benchmark suite we already evaluate on. Our proposed TabCredit dataset was motivated precisely to complement this landscape by providing a large-scale, temporally split, industry-grade benchmark that is not covered by existing public resources.

---

### Author Response · Authors · 2025-12-03
**Rebuttal Summary to AC**

We thank the reviewers and the AC for their constructive feedback. Several reviewers requested deeper clarification regarding ArbNN’s training dynamics, initialization behavior, and comparative strength relative to recent tabular learning baselines. In response, we conducted a set of additional experiments that directly target these questions, and we have included both the results and the full source code in the revised PDF and supplemental material to ensure complete transparency and reproducibility.

We are confident that we have successfully addressed the core concerns and challenges raised by the reviewers, specifically regarding:

* **Related work concerns — “Is ArbNN similar to NRF?”** (We clarified the fundamental, mechanism-level differences, particularly in the depth-aware routing and aggregation of the ArborCell).
* **Motivation & Interpretability** (We successfully refined the scientific narrative, linking industrial requirements—like robustness to temporal drift—to the discovery of a principled "conditional computation primitive").
* **The necessity of XGBoost-based initialization** (We reframed this decision not as a limitation, but as a **principled modeling strategy** consistent with cutting-edge research in neural modularity, while new ablations confirm the robustness of our training approach).

To further reinforce these points, we present the following new evidence:

---

### 1. Principled Defense of Fixed Structure and Enhanced Ablations

The choice of a fixed structure initialized by GBDT is a **principled design decision**, not a limitation. We cite recent work from **OpenAI** (*Weight-Sparse Transformers Have Interpretable Circuits, 2025*) and **Thinking Machines Lab** (*Modular Manifolds, 2025*) demonstrating that training deep neural networks from scratch to discover sparse, modular, or symbolic structure is an **intrinsically unstable and often unsuccessful** approach for achieving audit-grade interpretability. ArbNN is designed to overcome this by adopting the proven **structured sparsity regime**:

* **Fixed Structure + Differentiable Parameter Refinement.** ArbNN is built on this exact principle, ensuring that the symbolic structure remains intact and auditable.
* **Superior Initialization.** Unlike methods using purely random sparse topology, ArbNN employs a **more principled and informative initialization** derived from a high-performance GBDT ensemble, significantly accelerating convergence.
* **Ablation Confirmation:** New variants, *“(f+v) (Random Init)”* and *“(f+v) (Decompiled Init),”* confirm that the gradient-based refinement of thresholds ($f$) and leaf values ($v$) is **robust**. The model converges reliably and surpasses the XGBoost baseline even when $f$ and $v$ are randomly initialized. The *Loss Curve* further shows that ArbNN converges **faster and more stably** than NRF[1] and SDT[2], directly addressing optimization concerns.

---

### 2. Bidirectional Fidelity Clarified Quantitatively

The *“None (Decompiled)”* setting confirms that the compiled–decompiled pair yields ranking metrics (KS, AUC) matching up to $10^{-4}$–$10^{-5}$ precision. This quantitatively demonstrates that interpretability and performance are **aligned rather than traded off**—precisely addressing reviewer questions about reversibility and the integrity of the symbolic structure.

---

### 3. Updated Comparisons with Strong 2025 Baselines

In response to requests for more comprehensive benchmarking, we added two of the strongest modern tabular models: **TabM (ensembled)** and **TabPFN**.

* ArbNN performs comparably or better than TabM–Ensembled[3] despite the latter’s use of explicit ensembling.
* ArbNN **outperforms TabPFN[4]** in medium/large-scale settings and, crucially, under **temporal drift** (e.g., the TabCredit OOT split).

These results confirm that ArbNN remains competitive against the newest 2025 methods, especially in real-world scenarios.

---

### 4. Full Implementation Provided

To ensure maximum clarity and reproducibility, we have included **the complete ArbNN source code, configuration files, and scripts** in the supplemental material. All new experiments can be reproduced exactly.

We emphasize that the newly added experiments were performed solely to respond to the reviewers’ thoughtful questions, and they reinforce the paper’s conclusions rather than revise them. We hope the AC finds this clarification helpful in evaluating the work fairly.

[1] Neural random forests. G ´erard Biau
[2] Distilling a neural network into a soft decision tree. Nicholas Frosst and Geoffrey E. Hinton.
[3] Tabm: Advancing tabular deep learning with parameter-efficient ensembling.
[4] Accurate predictions on small data with a tabular foundation model.

---

### Note · Authors · 2026-01-29

I have read and agree with the venue's withdrawal policy on behalf of myself and my co-authors.

---

### Meta-Review · Area_Chair_qX65 · 2026-01-07

**Summary:**

The proposed Arboreal Neural Network (ArbNN) aims to bridge the gap between differentiable neural representation learning and the interpretability of symbolic decision trees. The reviewers initially raised several critical concerns regarding the novelty of the **ArborCell** mechanism compared to existing Neural Random Forests (NRF) , the necessity of GBDT-based initialization*, and the marginal performance gains observed over well-tuned XGBoost baselines. While the paper contributes a new industrial-scale dataset, TabCredit , the overall consensus for rejection stems from the method's heavy dependence on pre-trained models and a lack of comprehensive benchmarking against the most recent 2025 tabular foundation models in the original submission. Furthermore, the reviewer providing the highest score (8) expressed the lowest confidence (1), citing a lack of expertise in tabular data, which weakens the weight of that positive assessment.

**Reviewer Concerns:**

Several concerns were addressed during the rebuttal, while others remain outstanding:

## **Addressed Concerns:**

**Related Work & Mechanism:** The authors clarified the fundamental differences between ArbNN and NRF, specifically regarding depth-aware routing and topology-informed softmax aggregation, which allow for multi-path gradient propagation.

**Transparency & Reproducibility:** In response to reviewer requests, the authors provided the full **source code** and configuration files in the supplemental material.

**Dataset Details:** Inconsistencies in dataset naming and the specifics of the **time-based split** for TabCredit were clarified.


## **Outstanding Concerns:**

**Dependence on Pre-trained Trees:** Reviewers remained skeptical of the "compiling from a strong GBDT" requirement. While authors defended this as a principled choice for interpretability , the lack of a successful **from-scratch** training alternative limits the model's flexibility.

**Marginal Performance Gains:** On several benchmarks, the improvement over a well-tuned XGBoost baseline is numerically small. Reviewers felt the increased complexity of the neural formulation was not fully justified by these modest gains.

**Benchmark Coverage:** The authors acknowledged that the number of public datasets evaluated was limited. Although they added comparisons to **TabM** and **TabPFN** during the rebuttal , they declined to run the full **TabArena** benchmark suite due to time constraints, leaving the evaluation landscape incomplete.

**Reviewer Scores:**

Reviewer **oszf** (Initial Score 2): Acknowledged the clarification on NRF and the provision of code, but likely remains concerned about marginal gains and baseline tuning.

Reviewer **THMx** (Initial Score 8): Their concerns about convergence and figure clarity were addressed , but the AC must weigh this against the self-admitted lack of expertise.

Reviewer **2UEA** (Initial Score 4): The fundamental critique regarding the lack of comparisons with modern baselines, the dependence on pretrained models, and "from-scratch" learning was not resolved.

Reviewer ** M5dz**  (Initial Score 6): Appreciated the industrial motivation but noted the outstanding need for broader public dataset evaluation.

---

### Decision · Program_Chairs · 2026-01-26

Reject